# High-Dimensional Gaussian Process Bandits

**Josip Djolonga**
ETH Zürich
josipd@ethz.ch

**Andreas Krause**
ETH Zürich
krausea@ethz.ch

**Volkan Cevher**
EPFL
volkan.cevher@epfl.ch

## Abstract

Many applications in machine learning require optimizing unknown functions defined over a high-dimensional space from noisy samples that are expensive to obtain. We address this notoriously hard challenge, under the assumptions that the function varies only along some low-dimensional subspace and is smooth (i.e., it has a low norm in a Reproducible Kernel Hilbert Space). In particular, we present the SI-BO algorithm, which leverages recent low-rank matrix recovery techniques to learn the underlying subspace of the unknown function and applies Gaussian Process Upper Confidence sampling for optimization of the function. We carefully calibrate the exploration–exploitation tradeoff by allocating the sampling budget to subspace estimation and function optimization, and obtain the first subexponential cumulative regret bounds and convergence rates for Bayesian optimization in high-dimensions under noisy observations. Numerical results demonstrate the effectiveness of our approach in difficult scenarios.

## 1   Introduction

The optimization of non-linear functions whose evaluation may be noisy and expensive is a challenge that has important applications in sciences and engineering. One approach to this notoriously hard problem takes a Bayesian perspective, which uses the predictive uncertainty in order to trade exploration (gathering data for reducing model uncertainty) and exploitation (focusing sampling near likely optima), and is often called Bayesian Optimization (BO). Modern BO algorithms are quite successful, surpassing even human experts in learning tasks: e.g., gait control for the SONY AIBO, convolutional neural networks, structural SVMs, and Latent Dirichlet Allocation [1, 2, 3].

Unfortunately, the theoretical efficiency of these methods depends exponentially on the—often high—dimension of the domain over which the function is defined. A way to circumvent this "curse of dimensionality" is to make the assumption that only a small number of the dimensions actually matter. For example, the cost function of neural networks effectively varies only along a few dimensions [2]. This idea has been also at the root of nonparametric regression approaches [4, 5, 6, 7].

To this end, we propose an algorithm that learns a low dimensional, *not necessarily axis-aligned*, subspace and then applies Bayesian optimization on this estimated subspace. In particular, our SI-BO approach combines low-rank matrix recovery with Gaussian Process Upper Confidence Bound sampling in a carefully calibrated manner. We theoretically analyze its performance, and prove bounds on its cumulative regret. *To the best of our knowledge, we prove the first subexponential bounds for Bayesian optimization in high dimensions under noisy observations.* In contrast to existing approaches, which have an exponential dependence on the ambient dimension, our bounds have in fact polynomial dependence on the dimension. Moreover, our performance guarantees depend explicitly on what we could have achieved if we had known the subspace in advance.

**Previous work.**   Exploration–exploitation tradeoffs were originally studied in the context of finite multi-armed bandits [8]. Since then, results have been obtained for continuous domains, starting with the linear [9] and Lipschitz-continuous cases [10, 11]. A more recent algorithm that enjoys theoretical bounds for functions sampled from a Gaussian Process (GP), or belong to some Repro-

ducible Kernel Hilbert Space (RKHS) is GP-UCB [12]. The use of GPs to negotiate exploration–exploitation tradeoffs originated in the areas of response surface and Bayesian optimization, for which there are a number of approaches (cf., [13]), perhaps most notably the Expected Improvement [14] approach, which has recently received theoretical justification [15], albeit only in the noise-free setting.

Bandit algorithms that exploit low-dimensional structure of the function appeared first for the linear setting, where under sparsity assumptions one can obtain bounds, which depend only weakly on the ambient dimension [16, 17]. In [18] the more general case of functions sampled from a GP under the same sparsity assumptions was considered. The idea of random projections to BO has been recently introduced [19]. They provide bounds on the simple regret under noiseless observations, while we also analyze the cumulative regret and allow noisy observations. Also, unless the low-dimensional space is of dimension 1, our bounds on the simple regret improve on theirs. In [7] the authors approximate functions that live on low-dimensional subspaces using low-rank recovery and analysis techniques. While providing uniform approximation guarantees, their approach is not tailored towards exploration–exploitation tradeoffs, and does not achieve sublinear cumulative regret. In [20] the stochastic and adversarial cases for axis-aligned Hölder continuous functions are considered.

Our specific contributions in this paper can be summarized as follows:

- We introduce the SI-BO algorithm for Bayesian bandit optimization in high dimensions, admitting a large family of kernel functions. Our algorithm is a natural but non-trivial fusion of modern low-rank subspace approximation tools with GP optimization methods.
- We derive theoretical guarantees on SI-BO's cumulative and simple regret in high-dimensions *with noise*. To the best of our knowledge, these are the first theoretical results on the sample complexity and regret rates that are subexponential in the ambient dimension.
- We provide experimental results on synthetic data and classical benchmarks.

## 2 Background and Problem Statement

**Goal.** In plain words, we wish to sequentially optimize a bounded function over a compact, convex subset $D \subset \mathbb{R}^d$. Without loss of generality, we denote the function by $f : D \to [0, 1]$ and let $\mathbf{x}^*$ be a maximizer. The algorithm proceeds in a total of $T$ rounds. In each round $t$, it asks an oracle for the function value at some point $\mathbf{x}_t$ and it receives back the value $f(\mathbf{x}_t)$, possibly corrupted by noise. Our goal is to choose points such that their values are close to the optimum $f(\mathbf{x}^*)$.

As performance metric, we consider the regret, which tells us how much better we could have done in round $t$ had we known $\mathbf{x}^*$, or formally $r_t = f(\mathbf{x}^*) - f(\mathbf{x}_t)$. In many applications, such as recommender systems, robotic control, etc., we care about the quality of the points chosen at every time step $t$. Hence, a natural quantity to consider is the *cumulative regret* defined as $R_T = \sum_{t=1}^{T} r_t$. One can also consider the *simple regret*, defined as $S_T = \min_{t=1}^{T} r_t$, measuring the quality of the best solution found so far. We will give bounds on the more challenging notion of cumulative regret, which also bounds the simple regret via $S_T \leq R_T/T$.

**Low-dimensional functions in high-dimensions.** Unfortunately, our problem cannot be tractably solved without further assumptions on the properties of the function $f$. What is worse is that the usual compact support and smoothness assumptions cannot achieve much: the minimax lower bound on the sample complexity is exponential in $d$ [21, 6, 7]. We hence assume that the function effectively varies only along a small number of true active dimensions: i.e., the function lives on a $k \ll d$-dimensional subspace. Typically, $k$ or an upper bound on $k$ is assumed known [4, 5, 7, 6].

Formally, we suppose that there exists some function $g : \mathbb{R}^k \to [0, 1]$ and a matrix $A \in \mathbb{R}^{k \times d}$ with orthogonal rows so that $f(\mathbf{x}) = g(A\mathbf{x})$. We will additionally assume that $g \in \mathcal{C}^2$, which is necessary to bound the errors from the linear approximation that we will make. Further, w.l.o.g., we assume that $D = \mathbb{B}^d(1+\bar{\varepsilon})$ for some $\bar{\varepsilon} > 0$, where we define $\mathbb{B}^d(r)$ to be the closed ball around 0 of radius $r$ in $\mathbb{R}^d$.[1] To be able to recover the subspace we also need the condition that $g$ has Lipschitz continuous second order derivatives and a full rank Hessian at $\mathbf{0}$, which is satisfied for many functions [7].

**Smooth, low-complexity functions.** In addition to the low-dimensional subspace assumption, we also assume that $g$ is smooth. One way to encode our prior is to assume that the function $g$ resides in

**Algorithm 1** The SI-BO algorithm
___
**Require:** $m_X, m_\Phi, \lambda, \varepsilon, k$, oracle for $f$, kernel $\kappa$
  $\mathcal{C} \leftarrow m_X$ samples uniformly from $\mathbb{S}^{d-1}$
  **for** $i \leftarrow 1$ to $m_X$ **do**
    $\Phi_i \leftarrow m_\Phi$ samples uniformly from $\{\pm 1/\sqrt{m_\Phi}\}^k$
  $\mathbf{y} \leftarrow$ compute using Equation 1
  $\hat{X}_{DS} \leftarrow$ Dantzig Selector using $\mathbf{y}$, see Equation 2 and compute the SVD $\hat{X}_{DS} = \hat{U}\hat{\Sigma}\hat{V}^T$
  $\hat{A} \leftarrow \hat{U}^{(k)}$ // *Principal k vectors of* $\hat{U}$, $\mathcal{D} \leftarrow$ all $(\hat{A}\mathbf{x}, y)$ pairs queried so far
  Use GP inference to obtain $\mu_1(\cdot), \sigma_1(\cdot)$.
  **for** $t \leftarrow 1$ to $T - m_X(m_\Phi + 1)$ **do**
    $\mathbf{z}_t \leftarrow \arg\max_{\mathbf{z}} \mu_t(\mathbf{z}) + \beta_t^{1/2}\sigma_t(\mathbf{z})$, $y_t \leftarrow f(\hat{A}^T\mathbf{z}_t) +$ noise , $\mathcal{D}.\text{add}(\mathbf{z}_t, y_t)$
___

a Reproducing Kernel Hilbert Space (RKHS; cf., [23]), which allows us to quantify $g$'s complexity via its norm $\|g\|_{\mathcal{H}_\kappa}$. The RKHS for some positive semidefinite kernel $\kappa(\cdot, \cdot)$ can be constructed by completing the set of functions $\sum_{i=1}^n \alpha_i \kappa(\mathbf{x}_i, \cdot)$ under a suitable inner product. In this work, we use isotropic kernels, i.e., those that depend only on the distance between points, since the problem is rotation invariant and we can only recover $A$ up to some rotation.

Here is a final summary of our problem and its underlying assumptions:

1. We wish to maximize $f : \mathbb{B}^d(1 + \bar{\varepsilon}) \to [0, 1]$, where $f(\mathbf{x}) = g(A\mathbf{x})$ for some matrix $A \in \mathbb{R}^{k \times d}$ with orthogonal rows and $g$ belongs to some RKHS $\mathcal{H}_\kappa$.
2. The kernel $\kappa$ is isotropic $\kappa(\mathbf{x}, \mathbf{x}') = \kappa'(\mathbf{x} - \mathbf{x}') = \kappa''(\|\mathbf{x} - \mathbf{x}'\|_2)$ and $\kappa'$ is continuous, integrable and with a Fourier transform $\mathcal{F}\kappa'$ that is isotropic and radially non-increasing.[2]
3. The function $g$ has Lipschitz continuous $2^{\text{nd}}$-order derivatives and a full rank Hessian at $\mathbf{0}$.
4. The function $g$ is $\mathcal{C}^2$ on a compact support and $\max_{|\beta| \leq 2} \|D^\beta g\|_\infty \leq C_2$ for some $C_2 > 0$.
5. The oracle noise is Gaussian with zero mean with a known variance $\sigma^2$.

## 3 The SI-BO Algorithm

The SI-BO algorithm performs two separate exploration and exploitation stages: (1) subspace identification (SI), i.e. estimating the subspace on which the function is supported, and then (2) Bayesian optimization (BO), in order to optimize the function on the learned subspace. A key challenge here is to carefully allocate samples between these phases.

We first give a detailed outline for SI-BO in Alg. 1, deferring its theoretical analysis to Section 4. Given the (noisy) oracle for $f$, we first evaluate the function at several suitably chosen points and then use a low-rank recovery algorithm to compute a matrix $\hat{A}$ that spans a subspace well aligned with the one generated by the true matrix $A$. Once we have computed $\hat{A}$, similarly to [22, 7], we define the function which we optimize as $\hat{g}(\mathbf{z}) = f(\hat{A}^T\mathbf{z}) = g(A\hat{A}^T\mathbf{z})$. Thus, we effectively work with an approximation $\hat{f}$ to $f$ given by $\hat{f}(\mathbf{x}) = \hat{g}(\hat{A}\mathbf{x}) = g(A\hat{A}^T\hat{A}\mathbf{x})$. With the approximation at hand, we apply BO, in particular the GP-UCB algorithm, on $\hat{g}$ for the remaining steps.

**Subspace Learning.** We learn $A$ using the approach from [7], which reduces the learning problem to that of low rank matrix recovery. We construct a set of $m_X$ points $\mathcal{C} = [\xi_1, \cdots, \xi_{m_X}]$, which we call sampling centers, and consider the matrix $X$ of gradients at those points $X = [\nabla f(\xi_1), \cdots, \nabla f(\xi_{m_X})]$. Using the chain rule, we have $X = A^T[\nabla g(A\xi_1), \cdots, \nabla g(A\xi_{m_X})]$. Because $A$ is a matrix of size $k \times d$ it follows that the rank of $X$ is at most $k$. This suggests that using low-rank approximation techniques, one may be able to (up to rotation) infer $A$ from $X$.

Given that we have no access to the gradients of $f$ directly, we approximate $X$ using a linearization of $f$. Consider a fixed sampling center $\xi$. If we make a linear approximation with step size $\varepsilon$ to the directional derivative at center $\xi$ in direction $\varphi$ then, by Taylor's theorem, for a suitable $\zeta(\mathbf{x}, \varepsilon, \varphi)$:

$$\langle \varphi, A^T\nabla g(A\xi) \rangle = \frac{1}{\varepsilon}(f(\xi + \varepsilon\varphi) - f(\xi)) - \underbrace{\frac{\varepsilon}{2}\varphi^T\nabla^2 f(\zeta)\varphi}_{E(\xi, \varepsilon, \varphi)}.$$

___
[2]This is the same assumption as in [15]. Radially non-increasing means that if $\|w\| \leq \|w'\|$ then $\mathcal{F}\kappa'(w) \geq \mathcal{F}\kappa'(w')$. Note that this is satisfied by the RBF and Matèrn kernels.

Thus, sampling the finite difference $f(\xi + \varepsilon\varphi) - f(\xi)$ provides (up to the curvature error $E(\xi, \varepsilon, \varphi)$, and sampling noise) information about the one-dimensional subspace spanned by $A^T \nabla g(A\xi)$. To estimate it accurately, we must observe multiple directions $\varphi$. Further, to infer the full $k$-dimensional subspace $A$, we need to consider at least $m_X \geq k$ centers. Consequently, for each center $\xi_i$, we define a set of $m_\Phi$ directions and arrange them in a total of $m_\Phi$ matrices $\Phi_i = [\varphi_{i,1}, \varphi_{i,2}, \cdots, \varphi_{i,m_X}] \in R^{d \times m_X}$. We can now define the following linear system:

$$\mathbf{y} = \mathcal{A}(X) + \mathbf{e} + \mathbf{z}, \quad y_i = \frac{1}{\varepsilon} \sum_{j=1}^{m_X} (f(\xi_j + \varepsilon\varphi_{i,j}) - f(\xi_j)), \tag{1}$$

where the linear operator $\mathcal{A}$ is defined as $\mathcal{A}(X)_i = \mathrm{tr}(\Phi_i^T X)$, the curvature errors have been accumulated in $\mathbf{e}$ and the noise has been put in the vector $\mathbf{z}$ which is distributed as $z_i \sim \mathcal{N}(0, 2m_X\sigma^2/\varepsilon)$.

Given the structure of the problem, we can make use of several low-rank recovery algorithms. For concreteness, we choose the Dantzig Selector (DS, [24]), which recovers low rank matrices via

$$\underset{M}{\text{minimize}} \ \|M\|_* \quad \text{subject to} \quad \|\mathcal{A}^*(\underbrace{\mathbf{y} - \mathcal{A}(M)}_{\text{residual}})\| \leq \lambda, \tag{2}$$

where $\|\cdot\|_*$ is the nuclear norm and $\|\cdot\|$ is the spectral norm. The DS will successfully recover a matrix $\hat{X}$ close to the true solution in the Frobenius norm and moreover this distance decreases linearly with $\lambda$. As shown in [7], choosing the centers $\mathcal{C}$ uniformly at random from the unit sphere $\mathbb{S}^{d-1}$, choosing each direction vector uniformly at random from $\{\pm 1/\sqrt{m_\Phi}\}^k$, and—in the case of noisy observations, resampling $f$ repeatedly—suffices to obtain an accurate $\hat{X}$ w.h.p., as long as $m_\Phi$ and $m_X$ are sufficiently large. The precise choices of these quantities are analyzed in Section 4.

Finally, we extract the matrix $\hat{A}$ from the SVD of $\hat{X}$, by taking its top $k$ left singular vectors. Because the DS will find a matrix $\hat{X}$ close to $X$, due to a result by Wedin [25] we know that the learned subspace will be close to the true one.

**Optimizing $\hat{g}$.** Once we have an approximate $\hat{A}$, we optimize the function $\hat{g}(\mathbf{z}) = f(\hat{A}^T \mathbf{z})$ on the low-dimensional domain $\mathcal{Z} = \mathbb{B}^k(1 + \bar{\varepsilon})$. Concretely, we use GP-UCB [12], because it exhibits state of the art empirical performance, and enjoys strong theoretical bounds for the cumulative regret. It requires that $\hat{g}$ belongs to the RKHS and the noise when conditioned on the history is zero-mean and almost surely bounded by some $\hat{\sigma}$. Section 4 shows that this is indeed true with high probability.

In order to trade exploration and exploitation, the GP-UCB algorithm computes, for each point $\mathbf{z}$, a score that combines the predictive mean that we have inferred for that point with its variance, which quantifies the uncertainty in our estimate. They are combined linearly with a time-dependent weighting factor $\beta_t$ in the following surrogate function

$$\mathrm{ucb}(\mathbf{z}) = \mu_t(\mathbf{z}) + \beta_t^{1/2}\sigma_t(\mathbf{z}) \tag{3}$$

for a suitably chosen $\beta_t = 2B + 300\gamma_t \log^3(t/\delta)$. Here, $B$ is an upper bound on the squared RKHS norm of the function that we optimize, $\delta$ is an upper bound on the failure probability and $\gamma_t$ depends on the kernel [12]: cf., Section 4.[3] The algorithm then greedily maximizes the ucb score above.

Note that finding the maximum of this non-convex and in general multi-modal function, while considered to be cheaper than evaluating $f$ at a new point, is by itself a hard problem and it is usually approached by either sampling on a grid in the domain, or using some global Lipschitz optimizer [13]. Hence, by reducing the dimension of the domain $\mathcal{Z}$ over which we have to optimize, our algorithm has the additional benefit that this process can be performed more efficiently.

**Handling the noise.** The last ingredient that we need is theory on how to pick $\hat{\sigma}$ so that it bounds the noise during the execution of GP-UCB w.h.p., and how to select $\lambda$ in (2) so that the true matrix $X$ is feasible in the DS. Due to the fast decay of the tails of the Gaussian distribution we can pick $\hat{\sigma} = \left(2\log\frac{1}{\delta} + 2\log T + \log\frac{1}{2\pi}\right)^{1/2}\sigma$, where $T$ is the number of GP-UCB iterations and $\sigma^2$ is the variance of the noise. Then the noise will be trapped in $[-\hat{\sigma}, \hat{\sigma}]$ with probability at least $1 - \delta$.

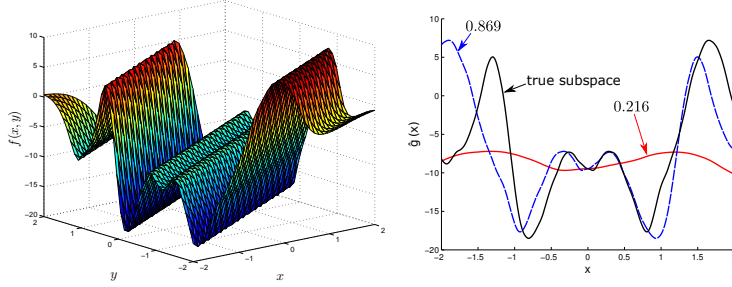

Figure 1: A 2-dimensional function $f(x, y)$ varying along a 1-dimensional subspace and its projections on different subspaces. The numbers are the respective cosine distances.

The analysis on $\lambda$ comes from [7]. They bound $\|\mathcal{A}^*(\mathbf{e} + \mathbf{z})\|$ using the assumption that the second order derivatives are bounded and, as shown in [24], because $\mathbf{z}$ has a Gaussian distribution,

$$\|\mathcal{A}^*(\mathbf{e} + \mathbf{z})\| \le 1.2 \left( \frac{C_2 \varepsilon d m_X k^2}{2\sqrt{m_\Phi}} + \frac{5\sqrt{m_X m_\Phi}\sigma}{\varepsilon} \right) \tag{4}$$

If there is no noise it still holds by setting $\sigma = 0$. This bound, intuitively, relates the approximation quality $\lambda$ of the subspace to the quantities $m_\Phi$, $m_X$ as well as the step size $\varepsilon$.

## 4 Theoretical Analysis

**Overview.** A crucial choice in our algorithm is how to allocate samples (by choosing $m_\Phi$ and $m_X$ appropriately) to the tasks of subspace learning and function optimization. We now analyze both phases, and determine how to split the queries in order to optimize the cumulative regret bounds.

Let us first consider the regret incurred in the second phase, in the ideal (but unrealistic) case that the subspace is estimated exactly (i.e., $\hat{A} = A$). This question was answered recently in [12], where it is proven that it is bounded by $\mathcal{O}^*(\sqrt{T}(B\sqrt{\gamma_t} + \gamma_t))$ [4]. Hereby, the quantity $\gamma_T$ is defined as

$$\gamma_T = \max_{S \subseteq D, |S| = T} H(\mathbf{y}_S) - H(\mathbf{y}_S | f),$$

where $\mathbf{y}_S$ are the values of $f$ at the points in $S$, corrupted by Gaussian noise, and $H(\cdot)$ is the entropy. It quantifies the maximal gain in information that we can obtain about $f$ by picking a set of $T$ points. In [12] sublinear bounds for $\gamma_T$ have been computed for several popular kernels. For example, for the RBF kernel in $k$ dimensions, $\gamma_T = O\big((\log T)^{k+1}\big)$. Further, $B$ is a bound on the squared norm $\|g\|^2_{\mathcal{H}_\kappa}$ of $g$ w.r.t. kernel $\kappa$. Note that generally $\gamma_T$ grows exponentially with $k$, rendering the application of GP-UCB directly to the high-dimensional problem intractable.

What happens if the subspace $\hat{A}$ is estimated incorrectly? Fortunately, w.h.p. the estimated function $\hat{g}$ still remains in the RKHS associated with kernel $\kappa$. However, the norm $\|\hat{g}\|_{\mathcal{H}_\kappa}$ may increase, and consequently may the regret. Moreover, the considered $\hat{f}$ disagrees with the true $f$, and consequently additional regret per sample may be incurred by $\eta = \|\hat{f} - f\|_\infty$. As an illustration of the effect of misestimated subspaces see Figure 1. We can observe that subspaces far from the true one stretch the function more, thus increasing its RKHS norm.

We now state a general result that formalizes these insights by bounding the cumulative regret in terms of the samples allocated to subspace learning, and the subspace approximation quality.

**Lemma 1** *Assume that we spend $0 < n \le T$ samples to learn the subspace such that $\|f - \hat{f}\|_\infty \le \eta$, $\|\hat{g}\| \le B$ and the error is bounded by $\hat{\sigma}$, each w.p. at least $1 - \delta/4$. If we run the GP-UCB algorithm for the remaining $T - n$ steps with the suggested $\hat{\sigma}$ and $\delta/4$, then the following bound on the cumulative regret holds w.p. at least $1 - \delta$*

$$R_T \le n + \underbrace{\eta T}_{\text{approx. error}} + \underbrace{\mathcal{O}^*(\sqrt{T}(B\sqrt{\gamma_t} + \gamma_t))}_{R_{UCB}(T, \hat{g}, \kappa)}$$

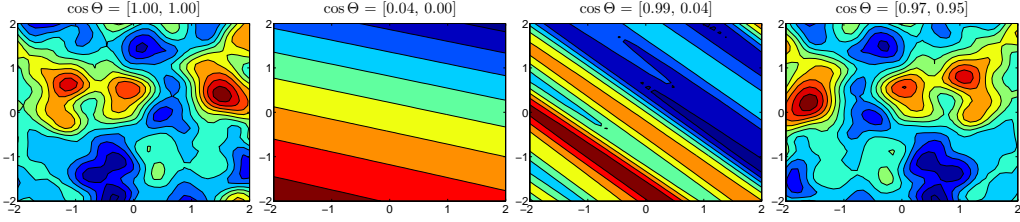

Figure 2: Approximations $\hat{g}$ resulting from differently aligned subspaces. Note that inaccurate estimation (the middle two cases) can wildly distort the objective.

where $R_{UCB}(T, \hat{g}, \kappa)$ is the regret of **GP-UCB** when run for $T$ steps using $\hat{g}$ and kernel $\kappa$ [5].

Lemma 1 breaks down the regret in terms of the approximation error incurred by subspace-misestimation, and the optimization error incurred by the resulting increased complexity $\|\hat{g}\|^2_{\mathcal{H}_\kappa} \leq B$. We now analyze these effects, and then prove our main regret bounds.

**Effects of Subspace Alignment.** A notion that will prove to be very helpful for analyzing both, the approximation precision $\eta$ and the norm of $\hat{g}$, is the set of angles between the subspaces that are defined by $A$ and $\hat{A}$. The following definition [26] makes this notion precise.

**Definition 2** *Let* $A, \hat{A} \in \mathbb{R}^{k \times d}$ *be two matrices with orthogonal rows so that* $AA^T = \hat{A}\hat{A}^T = I$. *We define the vector of cosines between the spanned subspaces* $\cos \Theta(A, \hat{A})$ *to be equal to the singular values of* $A\hat{A}^T$. *Analogously* $\sin \Theta(A, \hat{A})_i = (1 - \cos \Theta(A, \hat{A})_i^2)^{1/2}$.

Let us see how $\hat{A}$ affects $\hat{g}$. Because $\hat{g}(\mathbf{z}) = g(A\hat{A}^T \mathbf{z})$, the matrix $M = A\hat{A}^T$, which converts any point from its coordinates determined by $\hat{A}$ to the coordinates defined by $A$, will be of crucial importance. First, note that its singular values are cosines and are between $-1$ and $1$. This means that it can only shrink the vectors that we apply it to (possibly by different amounts in different directions). The effect on $\hat{g}$ is that it might only "see" a small part of the whole space, and its shape might be distorted, which in turn will increase its RKHS complexity (see Figure 2 for an illustration).

**Lemma 3** *If* $g \in \mathcal{H}_\kappa$ *for a kernel that is isotropic with a radially non-increasing Fourier transform and* $\hat{g}(x) = g(A\hat{A}^T x)$ *for some* $A, \hat{A}$ *with orthogonal rows, then for* $C = C_2 \sqrt{2k}(1 + \bar{\varepsilon})$,

$$\|f - \hat{f}\|_\infty \leq C \|\sin \Theta(A, \hat{A})\|_2 \quad \text{and} \quad \|\hat{g}\|^2_{\mathcal{H}_\kappa} \leq |\operatorname{prod} \cos \Theta(A, \hat{A})|^{-1} \|g\|^2_{\mathcal{H}_\kappa}. \quad (5)$$

Here, we use the notation $\operatorname{prod} \mathbf{x} = \prod_{i=1}^d x_i$ to denote the product of the elements of a vector. By decreasing the angles we tackle both issues: the approximation error $\eta = \|f - \hat{f}\|_\infty$ is reduced and the norm of $\hat{g}$ gets closer to the one of $g$. There is one nice interpretation of the product of the cosines. It is equal to the determinant of the matrix $M$. Hence, $\hat{g}$ will not be in the RKHS only if $M$ is rank deficient as dimensions are collapsed.

**Regret Bounds.** We now present our main bounds on the cumulative regret. In order to achieve sublinear regret, we need a way of controlling $\eta$ and $\|\hat{g}\|_{\mathcal{H}_\kappa}$. In the following, we show how this goal can be achieved. As it turns out, subspace learning is substantially harder in the case of noisy observations. Therefore, we focus on the easier, noise-free setting first.

*Noiseless Observations.* We should note that the theory behind **GP-UCB** still holds in the deterministic case, as it only requires the noise to be bounded a.s. by $\hat{\sigma}$. The following theorem guarantees that in this setting for non-linear kernels *we have a regret dominated by* **GP-UCB**, which is of order $\Omega^*(\sqrt{T}\gamma_T)$, as it is usually exponential in $k$.

**Theorem 4** *If the observations are noiseless we can pick* $m_x = \mathcal{O}(kd \log 1/\delta)$, $\varepsilon = \frac{1}{k^{2.25}d^{3/2}T^{1/2}}$ *and* $m_\varphi = \mathcal{O}(k^2 d \log 1/\delta)$ *so that with probability at least* $1 - \delta$ *we have the following*

$$R_T \leq \mathcal{O}(k^3 d^2 \log^2(1/\delta)) + 2 R_{UCB}(T, g, \kappa).$$

**Noisy Observations.** Equation 4 hints that the noise can have a dramatic effect in learning efficiency. As already mentioned, the DS gets better results as we decrease $\lambda$. In the noiseless case, it suffices to increase the number of directions $m_\Phi$ and decrease the step size $\varepsilon$ in estimating the finite differences. However, the second term in $\lambda$ can only be reduced by decreasing the variance $\sigma^2$.

As a result, each point that we evaluate is sampled $n$ times and we take as its value the average. Moreover, note that because the standard deviation decreases as $1/\sqrt{n}$, we have to resample at least $\varepsilon^{-2}$ times and this significantly increases the number of samples that we need. Nevertheless, we are able to obtain cumulative regret bounds (and thereby the first convergence guarantees and rates) for this setting, which *only polynomially depend on* $d$. Unfortunately, the dependence on $T$ is now weaker than those in the noiseless setting (Theorem 4), and the regret due to the subspace learning might dominate that of GP-UCB.

**Theorem 5** *If the observations are noisy, we can pick* $\varepsilon = \frac{1}{k^{2.25}d^{1.5}T^{1/5}}$ *and all other parameters as in the previous theorem. Moreover, we have to resample each point* $\mathcal{O}(\sigma^2 k^2 dT^{2/5} m_\Phi / \varepsilon^2)$ *times. Then, with probability at least* $1 - \delta$

$$R_T \leq \mathcal{O}\left(\sigma^2 k^{11.5} d^7 T^{4/5} \log^3(1/\delta)\right) + 2\, R_{UCB}(T, g, \kappa).$$

**Mismatch on the effective dimension** $k$. All models are imperfect in some sense and the structure of a general $f$ is impossible to identify unless we have further scientific evidence beyond the data. In our case, the assumption $f(\mathbf{x}) = g(A\mathbf{x})$ for some $k$ more or less takes the weakest form for indicating our hope that BO can succeed from a sub-exponential sample size. In general, we must tune $k$ in a degree to reflect the anticipated complexity in the learning problem. Fortunately, all the guarantees are preserved if we assume a $k > k_{\text{true}}$, for some true synthetic model, where $f(\mathbf{x}) = g(A\mathbf{x})$ holds. Underfitting $k$ leads to additional errors that are well-controlled in low-rank subspace estimation [24]. The impact of under fitting in our setting is left for future work.

## 5 Experiments

The main intent of our experiments is to provide a proof of concept, confirming that SI-BO not just in theory provides the first subexponential regret bounds, but also empirically obtains low average regret for Bayesian optimization in high dimensions.

**Baselines.** We compare SI-BO against the following baseline approaches:

- RandomS-UCB, which runs GP-UCB on a random subspace.
- RandomH-UCB, which runs GP-UCB on the high-dimensional space. At each iteration we pick 1000 points at random and choose the one with highest UCB score.
- Exact-UCB, which runs GP-UCB on the exact (but in practice unknown) subspace.

The $\beta_t$ parameter in the GP-UCB score was set as recommended in [12] for finite sets. To optimize the UCB score we sampled on a grid on the low dimensional subspace. For all of the measurements we have added Gaussian zero-mean noise with $\sigma = 0.01$.

**Data sets.** We carry out experiments in the following settings:

- *GP Samples.* We generate random 2-dimensional samples from a GP with Matèrn kernel with smoothness parameter $\nu = 5/2$, length scale $\ell = 1/2$ and signal variance $\sigma_f^2 = 1$. The samples are "hidden" in a random 2-dimensional subspace in 100 dimensions.
- *Gabòr Filters.* The second data set is inspired by experimental design in neuroscience [27]. The goal is to determine visual stimuli that maximally excite some neuron, which reacts to edges in the images. We consider the function $f(\mathbf{x}) = \exp(-(\theta^T \mathbf{x} - 1)^2)$, where $\theta$ is a Gabór filter of size $17 \times 17$ and the set of admissible signals is $[0, 1]^d$.

In the appendix we also include results for the Branin function, a classical optimization benchmark.

**Results.** The results are presented in Figure 3. We show the averages of 20 runs (10 runs for *GP-Posterior*) and the shaded areas represent the standard error around the mean. We show both the average regret and simple regret (i.e., suboptimality of the best solution found so far). We find that although SI-BO spends a total of $m_X(m_\Phi + 1)$ samples to learn the subspace and thus incurs

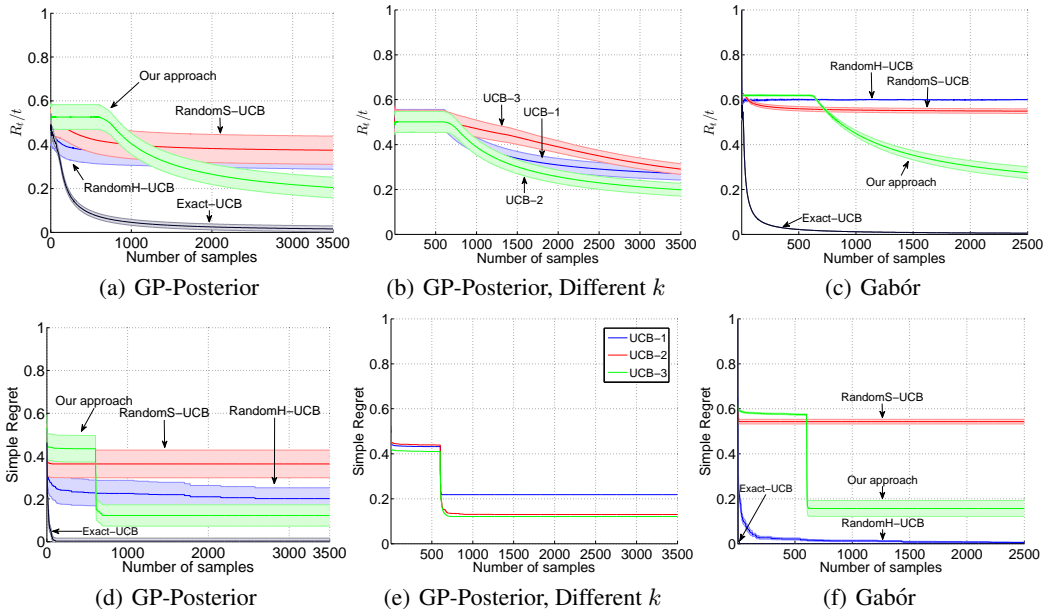

Figure 3: Performance comparison on different datasets. Our SI-BO approach outperforms the natural benchmarks in terms of cumulative regret, and competes well with the unrealistic Exact-UCB approach that knows the true subspace $A$.

much regret during this phase, learning the subspace pays off, both for average and simple regret, and SI-BO ultimately outperforms the baseline methods on both data sets. This demonstrates the value of accurate subspace estimation for Bayesian optimization in high dimensions.

**Mis-specified $k$.** What happens if we do not know the dimensionality $k$ of the low dimensional subspace? To test this, we experimented with the stability of SI-BO w.r.t. $k$. We sampled 2-dimensional GP-Posterior functions and ran SI-BO with $k$ set to $1, 2$ and $3$. From the figure above we can see that in this scenario SI-BO is relatively stable to this parameter mis-specification.

## 6 Conclusion

We have addressed the problem of optimizing high dimensional functions from noisy and expensive samples. We presented the SI-BO algorithm, which tackles this challenge under the assumption that the objective varies only along a low dimensional subspace, and has low norm in a suitable RKHS. By fusing modern techniques for low rank matrix recovery and Bayesian bandit optimization in a carefully calibrated manner, it addresses the exploration–exploitation dilemma, and enjoys cumulative regret bounds, which only polynomially depend on the ambient dimension. Our results hold for a wide family of RKHS's, including the popular RBF and Matèrn kernels. Our experiments on different data sets demonstrate that our approach outperforms natural benchmarks.

**Acknowledgments.** A. Krause acknowledges SNF 200021-137971, DARPA MSEE FA8650-11-1-7156, ERC StG 307036 and a Microsoft Faculty Fellowship. V. Cevher acknowledges MIRG-268398, ERC Future Proof, SNF 200021-132548, SNF 200021-146750, and SNF CRSII2-147633.

## Footnotes

[1]Our method method can be extended to any convex compact set, see Section 5.2 in [22].

[3]If the bound $B$ is not known beforehand then one can use a doubling trick.

[4] We have used the notation $\mathcal{O}^*(f) = \mathcal{O}(f \log f)$ to suppress the log factors. $\Omega^*(\cdot)$ is analogously defined.

[5] Because the noise parameter $\hat{\sigma}$ depends on $T$, we have to slightly change the bounds from [12] as we have a term of order $\mathcal{O}(\sqrt{\log T + \log(1/\delta)})$; c.f. supplementary material.

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
