[Supplementary Material]

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

## APPENDIX : SUPPLEMENTARY MATERIAL

## A    Effects of Subspace Alignment

There is an alternative view of looking at RKHS's using the Fourier Transform (FT) of the kernel. The following two lemmas can be found in [15][§3.1] where proofs and additional references can also be found. Note that because of the conditions on the kernel its FT is integrable and non-negative.

**Lemma 6** (Lemma 1 in [15]) *$\mathcal{H}_\kappa(\mathbb{R}^d)$ is the space of real continuous functions $f \in L^2(\mathbb{R}^d)$ whose norm*

$$\|f\|^2_{\mathcal{H}_\kappa(\mathbb{R}^d)} = \int_{\mathbb{R}^d} \frac{|\mathcal{F}f(\omega)|^2}{\mathcal{F}\kappa(\omega)} d\omega$$

*is finite, taking $0/0 = 0$.*

**Lemma 7** (Lemma 2 in [15]) *$\mathcal{H}_\kappa(S)$ is the space of functions $f = g|_S$ for some $g \in \mathcal{H}_\kappa(\mathbb{R}^d)$ with norm*

$$\|f\|_{\mathcal{H}_\kappa(S)} = \inf_{g|_S=f} \|g\|_{\mathcal{H}_\kappa(\mathbb{R}^d)}$$

*and there is a unique g minimizing this expression.*

We will need the following well known result, which can be interpreted as the multivariate version of the popular stretch lemma for the one dimensional case.

**Lemma 8** *Let $g \in L^2(\mathbb{R}^n)$ be a function with Fourier transform $\mathcal{F}g$. For any function $g'$ that has the form*

$$g'(\mathbf{z}) = g(M\mathbf{z})$$

*for some non-singular matrix $M$ we have that*

$$\mathcal{F}g'(\omega) = |\det M|^{-1} g(M^{-T}\omega)$$

**Proof**

$$\begin{aligned}
\mathcal{F}g'(\omega) &= \int_{\mathbb{R}^n} \exp(-2\pi i \mathbf{z}^T \omega) g'(\mathbf{z}) d\mathbf{z} \\
&= \int_{\mathbb{R}^n} \exp(-2\pi i \mathbf{z}^T \omega) g(M\mathbf{z}) d\mathbf{z} \\
&= |\det M|^{-1} \int_{\mathbb{R}^n} \exp(-2\pi i \mathbf{z}^T M^{-T} \omega) g(\mathbf{z}) d\mathbf{z} \\
&= |\det M|^{-1} \mathcal{F}g(M^{-T}\omega)
\end{aligned}$$

The third step followed from integration by substitution.                                    □

We can now show how the RKHS norm of the learned function relates to that of the true low-dimensional function.

**Lemma 9** *If $g \in \mathcal{H}(\mathcal{Z})$ and $\hat{g} : \hat{\mathcal{Z}} \to \mathbb{R}$ is defined as $\hat{g}(\mathbf{z}) = g(M\mathbf{z})$ for some non-singular $M$ that satisfies $0 < \|M\| \leq 1$ for some matrix norm compatible with the Eucledian norm $\ell_2$, then the following holds*

$$\|\hat{g}\|^2_{\mathcal{H}_k(\hat{\mathcal{Z}})} \leq \frac{1}{|\det M|} \|g\|^2_{\mathcal{H}_k(\mathcal{Z})}$$

**Proof** The proof is very simple and is similar to that of [15][*Lem. 4*]. If $\mathcal{Z} = \mathbb{R}^n$, then

$$
\begin{aligned}
\|\hat{g}\|^2_{\mathcal{H}_k(\mathbb{R}^n)} &\overset{(1)}{=} \int \frac{|\mathcal{F}\hat{g}(\omega)|^2}{\mathcal{F}k(\omega)} d\omega \\
&\overset{(2)}{=} |\det M|^{-2} \int \frac{|\mathcal{F}g(M^{-T}\omega)|^2}{\mathcal{F}k(\omega)} d\omega \\
&\overset{(3)}{=} |\det M|^{-1} \int \frac{|\mathcal{F}g(\omega)|^2}{\mathcal{F}k(M^T\omega)} d\omega \\
&\overset{(4)}{\leq} |\det M|^{-1} \int \frac{|\mathcal{F}g(\omega)|^2}{\mathcal{F}k(\omega)} d\omega \\
&\overset{(5)}{=} \frac{1}{|\det M|} \|g\|^2_{\mathcal{H}_k(\mathbb{R}^n)}
\end{aligned}
$$

The second equality follows from Lemma 8 and the third one from integration by substitution and the fact that $\det M^T = \det M$. Noticing that $0 < \|M\| = \|M^T\| \leq 1$ we can use the compatibility of $\|\cdot\|$ with the $\ell_2$ norm to conclude that $\|Mw\| \leq \|M\|\|w\|_2 \leq \|w\|_2$ Then, we apply the fact that $\mathcal{F}k$ is radially non-increasing to obtain (4). Finally, we use Lemma 6. If $\mathcal{Z} \subset \mathbb{R}^n$, let $g^H \in \mathcal{H}_k(\mathbb{R}^n)$ be the unique minumum norm extension of $g$ (Lemma 7). Then $\hat{g}^H(\mathbf{z}) = g^H(M\mathbf{z})$ agrees with $\hat{g}$ on $\hat{\mathcal{Z}}$ and we can use the above argument. $\qquad\square$

We can now finally prove the central result in this section.

**Proof** (Lemma 3) The first part of the lemma is already known, e.g. see Appendix F in [7]. The second conclusion follows from Lemma 9. $\qquad\square$

## B  Regret Bounds

### B.1  Almost Surely Bounded Error

Assume that the noise is $\mathcal{N}(0, \sigma^2)$ and let $e_i$ denote the noise at iteration $i$. We denote by $z$ a random variable distributed as $\mathcal{N}(0,1)$. The probability of having an error bigger than $\hat{\sigma} = t\sigma$ is equal to

$$
\begin{aligned}
\mathbb{P}\{\max_{i=1}^{T} |\varepsilon_i| > t\sigma\} &\leq T\mathbb{P}(|\varepsilon_1| > t\sigma) \\
&= T\mathbb{P}(|z| > t) \\
&\leq \frac{T}{\sqrt{2\pi}} \exp(-t^2/2)
\end{aligned}
$$

where we have used a standard Gaussian tail inequality. Hence if we pick

$$
\hat{\sigma} = (2\log T + 2\log \frac{1}{\delta} + \log \frac{1}{2\pi})^{1/2}\sigma
$$

the probability of observing noise outside $[-\hat{\sigma}, \hat{\sigma}]$ is at most $\delta$.

### B.2  Proof of Lemma 1

**Proof** The first $n$ samples are spent on learning the subspace and a trivial bound on the regret of these first steps is $n$. Because by assumption the maxima of $g$ and $\hat{g}$ differ by at most $\eta$ and GP-UCB$(T - n, \hat{g}, k)$ is the regret w.r.t. $\hat{g}$, at each step we will accrue at most $\eta$ additional regret. Hence, with probability at least $1 - \delta$ the total regret is bounded by

$$
\underbrace{n}_{\text{subs. learning}} + \underbrace{(T - n)\eta}_{\text{approx.}} + \text{GP-UCB}(T - n, \hat{g}, k)
$$

We now apply Theorem 3 from [12] to GP-UCB$(T, \hat{g}, k)$ and this completes the proof. $\qquad\square$

Because for non-linear functions $\mathsf{GP\text{-}UCB}(\cdot) = \Omega^*(\sqrt{T})$ if we can assure that $\eta = \mathcal{O}(1/\sqrt{T})$ and $n = \mathcal{O}^*(\sqrt{T})$, then the regret will be dominated by $\mathsf{GP\text{-}UCB}$.

Note that our $\hat{\sigma}$ is a function of $T$ and the bounds in [12] depend on $\hat{\sigma}$, which is treated as a constant. More precisely, in Theorem 3 in [12] we have to change

$$\frac{1}{\log(1 + \frac{1}{\sigma^2})} \quad \text{to} \quad \frac{1}{\log(1 + \frac{1}{(2\log T + 2\log\frac{1}{\delta} + \log\frac{1}{2\pi})\sigma^2})} \ .$$

Note that the function $1/\log(1+1/x)$ is bounded by $x+1$, so in the worst case we will be multiplying the cumulative regret bounds by a term of order $\mathcal{O}(\sqrt{\log T + \log(1/\delta)})$.

## B.3 Notation

Because there are plenty of variables and constants that we use in the proofs of the regret bounds, we provide their definitions in the table below.

| Symbol | Meaning |
|---|---|
| $X$ | Contains the derivatives at the centers |
| $\sigma_k$ | $k$-th largest singular value of $X$ |
| $\hat{X}$ | The matrix obtained from the DS |
| $\hat{X}^{(k)}$ | Closest rank $k$ matrix to $\hat{X}$ in $\|\cdot\|_F$ |
| $d$ | Ambient dimension |
| $k$ | The dimension of the subspace |
| $m_X$ | Number of centers |
| $m_\Phi$ | Number of directions |
| $\varepsilon$ | The step in the numerical gradient |
| $C_2$ | Bound on second derivatives of $g$ |
| $\alpha$ | The $k$-th singular value of $H^f$ |
| $\kappa$ | Bound on the RIP constant |
| $\eta$ | Bound on $\|f - \hat{f}\|_\infty$ |
| $\delta$ | Bound on the failure probability |
| $\lambda$ | Controls the feasibility region in the DS |

## B.4 Background

Before showing the regret bounds results we must first set the stage by stating some results on subspace learning. Many of these results have been already known from [7] and [22], but we extend them if necessary for our setting.

As we have already mentioned, the error of the DS decreases with $\lambda$ and this is stated precisely in the next lemma. The probability is taken over the $m_\Phi$ sampling directions.

**Lemma 10** (Corollary 1 in [7]) *If $X$ is feasible in the DS then the matrix $\hat{X}^{(k)}$ satisfies*

$$\|X - \hat{X}^{(k)}\|_F^2 \leq \tau^2 \leq 4k\lambda^2$$

*w.p. at least*

$$1 - \underbrace{2\exp(-m_\Phi q(\kappa) + 4k(d + m_X + 1)u(\kappa))}_{p_1(m_X, m_\Phi)}$$

*for some $0 \leq \kappa < \sqrt{2} - 1$ and $u(\kappa) = \log(\frac{36\sqrt{2}}{\kappa})$, $q(\kappa) = \frac{1}{144}(\kappa^2 - \frac{\kappa^3}{9})$.*

Due to a result by Wedin [25] we can show that if two matrices are close in the Frobenius norm, then the subspaces spanned from their left singular vectors are aligned.

**Lemma 11** *If $\|X - \hat{X}^{(k)}\|_F \leq \tau$ and we extract $\hat{A}$ from the top $k$ left singular vectors of $\hat{X}^{(k)}$ then*

$$\|\sin\Theta(A, \hat{A})\|_2 \leq \frac{\sqrt{2}}{\sigma_k - \tau}\tau$$

**Proof** Follows from the proof of Lemma 2 in [7] and Theorem I.5.5 in [26]. □

We will need the following technical lemma for one of the central results that we will show.

**Lemma 12** *Let*

$$X = \{\mathbf{x} \in (0,1]^n \mid \sum_{i=1}^n x_i = n - 1 + \mu\}$$

*for some $\mu \in (0,1]$. Then $\min_{\mathbf{x} \in X} \prod_{i=1}^n x_i = \mu$ and the minimum is achieved at points $\mathbf{x}$ which have one coordinate equal to $\mu$ and all other to 1.*

**Proof** First I will show that at the minimum there must exist some $i$ such that $x_i = \mu$. Assume the contrary. Then all of the elements are greater than $\mu$ (if at least one of them is smaller than $\mu$ then the equality can not be satisfied). Moreover we assume w.l.o.g. that they are ordered so that $\mu < \underbrace{\mu + m_1}_{x_1} \leq \cdots \leq \underbrace{\mu + m_n}_{x_n}$ and all $m_i$ are positive. We consider two cases

(i) Assume $\mu + m_1 + m_2 \leq 1$. Consider the modified vector $\hat{\mathbf{x}}$ where we set

$$\hat{x}_1 = \mu, \; \hat{x}_2 = \mu + m_1 + m_2, \; \hat{x}_i = x_i \text{ for } i > 2$$

The constraints are again satisfied, so this is a valid assignment. The value of this assignment is $A' = C\mu(\mu + m_1 + m_2)$ while the old one had a value of $A = C(\mu + m_1)(\mu + m_2)$. It is easy to check that $A' < A$ and this is a contradiction.

(ii) Assume $\mu + m_1 + m_2 = 1 + \beta$ for some $\beta > 0$. Note that $x_1 + x_2 = \mu + \mu + m_1 + m_2 = 1 + \beta + \mu \leq 2$. Hence $\beta + \mu \leq 1$ and we can create a new vector $\hat{\mathbf{x}}$ as

$$\hat{x}_1 = \beta + \mu, \; \hat{x}_2 = 1, \; \hat{x}_i = x_i \text{ for } i > 2$$

The new value is $A' = (2\mu + m_1 + m_2 - 1)C$ while the old one was $A = (\mu + m_1)(\mu + m_2)C$. By assumption $A' \geq A$ which is possible only if the following quadratic is not positive

$$q(\mu) = \mu^2 + \mu(m_1 + m_2 - 2) + (1 + m_1 m_2 - m_2 - m_1)$$

By finding the zeros of $q$ we see that $A' \geq A$ only if $\mu \in [1 - m_2, 1 - m_1]$. However if $\mu \geq 1 - m_2$, then $x_2 = \mu + m_2 \geq 1$. This is impossible, because it would imply that $x_2 = x_3 = \cdots = x_n = 1$. Then $\sum_{i=1}^n x_n = x_1 + n - 1 > \mu + n - 1$, which is a contradiction.

Hence we can conclude that at least one of the elements in the optimal solution has to be $\mu$. Then the set $X$ over which we optimize consists of points which have one element equal to $\mu$ and all other to 1. □

The following lemma will tell us how precise our solutions from the DS has to be in order to have a good approximation accuracy and have the norm of the $\hat{g}$ under control.

**Lemma 13** *If for some $F, \eta \in (0,1)$ the recovered matrix $\hat{X}^{(k)}$ satisfies*

$$\underbrace{\|X - \hat{X}^{(k)}\|_F}_{\tau} \leq \underbrace{\min\{\frac{1}{1 + \sqrt{\frac{2}{1-F}}} \frac{\eta}{\eta + 2C_2\sqrt{k}(1+\bar{\varepsilon})}\}}_{\Xi} \sigma_k$$

*then $\|f - \hat{f}\|_\infty \leq \eta$ and $\|\hat{g}\|^2_{\mathcal{H}_k} \leq \frac{1}{F}\|g\|^2_{\mathcal{H}_k}$.*

**Proof** The first claim follows directly from Lemmas 3 and 11. For brevity let us denote $\sin \Theta = (\lambda_1, \cdots, \lambda_k)$ and $\cos \Theta = (\gamma_1, \cdots, \gamma_k)$. From Lemma 11 it follows that

$$\sum_{i=1}^k \lambda_i^2 = k - \sum_{i=1}^k \gamma_i^2 \leq 2(\frac{\tau}{\sigma_k - \tau})^2$$

$$\Downarrow$$

$$\sum_{i=1}^k |\gamma_i| \overset{|\gamma_i| \leq 1}{\geq} \sum_{i=1}^k \gamma_i^2 \geq k - 2(\frac{\tau}{\sigma_k - \tau})^2$$

Assume that we want to assure that the minimal cosine is at least $C$ for some $C \in (0,1)$. This is obviously satisfied if

$$\sum_{i=1}^{k} |\gamma_i| \geq k - 2(\frac{\tau}{\sigma_k - \tau})^2 > k - 1 + C$$

because then

$$\min_i |\gamma_i| \geq 1 - 2(\frac{\tau}{\sigma_k - \tau})^2 > C$$

Which will be satisfied if

$$\frac{\tau}{\sigma_k - \tau} < \sqrt{\frac{1-C}{2}} \implies \tau < \frac{1}{1 + \sqrt{\frac{2}{1-C}}} \sigma_k$$

We can now apply Lemma 12 and Lemma 3 to obtain the final result. $\square$

The final ingredient necessary before showing the regret bounds is a lower bound on $\sigma_k$ which can be obtained from a concentration inequality on the sum of positive semi-definite matrices [28]. The bound that we use depends on the quantity $\alpha$, which is equal to the $k$-th largest singular value of the following "Hessian" matrix

$$H^f = \int_{\mathbb{S}^{d-1}} \nabla f(\mathbf{x}) \nabla f(\mathbf{x})^T d\mu,$$

where $\mu$ is the uniform measure on $\mathbb{S}^{d-1}$.

**Lemma 14** (Lemma 4.5 in [22]) *For any $\rho \in (0,1)$ the $k$-th singular value $\sigma_k$ of the matrix $X$ that we try to retrieve with the DS satisfies*

$$\sigma_k \geq \sqrt{(1-\rho)m_X \alpha}$$

*w.p at least* $1 - \underbrace{k \exp(-\frac{m_X \alpha \rho^2}{2kC_2^2})}_{p_2(m_X)}$.

Moreover, it has been proven in [7] that under the assumptions that we made for the Hessian of $f$ and the Lipschitz continuity of the second derivatives of $f$, $\alpha$ behaves as $\alpha = \Theta(1/d)$. We will use this fact when proving the cumulative regret bounds.

## B.5 Bounding The Failure Probability

Other than the noise failing to be bounded a.s., there is another possible source of failure – the subspace learning algorithm can fail to learn the subspace within the required accuracy. The following lemma precisely bounds the probability of that happening.

**Lemma 15** *The residual is bounded by $\lambda$, i.e. Equation 4 holds, with probability at least*

$$\underbrace{1 - \exp(-cm_X)}_{p_3(m_X)} \text{ for some } c > 0$$

**Proof** First appeared in [7] based on Lemma 1.1 in [24]. $\square$

**Lemma 16** *We can pick $m_X = \mathcal{O}(\frac{k}{\alpha} \log(1/\delta))$ and $m_\Phi = \mathcal{O}(\frac{kd}{\alpha} \log(1/\delta))$ so that the failure probability is at most $\delta$. Moreover, if $\alpha = \Theta(1/d)$ then we have a better dependence on $d$ and we can use $m_\Phi = \mathcal{O}(k^2 d \log(1/\delta))$.*

**Proof** The failure probability for the algorithm is at most $p_1(m_X, m_\Phi) + p_2(m_X) + p_3(m_X)$. From the previous lemma we can obviously achieve $p_3 = \delta/3$ if we have $m_X = \mathcal{O}(\log(1/\delta))$. Because only $m_X$ appears in $p_2$ if we pick (assuming $\delta < 3/k$)

$$m_X \geq \frac{4kC_2^2}{\alpha\rho^2} \log(3/\delta) \geq \frac{2kC_2^2}{\alpha\rho^2}(\log(3/\delta) + \log k)$$

then $p_2 \leq \delta/3$. Similarly, by taking a look at $p_1(m_X, m_\Phi)$ we see that by choosing $m_\Phi$ as

$$m_\Phi \geq \frac{1}{q(\kappa)} \Big[ 4k(d+1)u(\kappa) + \log(6/\delta) + 4ku(\kappa)\frac{4kC_2^2}{\alpha\rho^2} \log(3/\delta) \Big]$$

we also have $p_1 \leq \delta/3$. The special case for $\alpha = \Theta(1/d)$ follows easily as none of the terms above contain both $d$ and $\alpha$. $\square$

## B.6 Regret Bounds

The following lemma combines many of the results that we have stated so far and will be very useful in showing the regret bounds.

**Lemma 17** *If*

$$4k \, 1.2 \underbrace{\left( \frac{C_2 \varepsilon d m_X k^2}{2\sqrt{m_\Phi}} + \frac{5\sqrt{m_X m_\Phi}\sigma}{\varepsilon} \right)^2}_{\lambda^2} \leq \Xi^2 (1-\rho) m_X \alpha$$

*then*

$$\tau \leq \Xi \sigma_k \text{ w.p. at least } 1 - p_1(m_X, m_\Phi) - p_2(m_X) - p_3(m_\Phi)$$

**Proof** We have that with at least the claimed probability Lemmas 15, 14 and 10 hold, in which case

$$\tau \overset{\text{Lemma 10}}{\leq} 2\sqrt{k}\lambda \leq \Xi\sqrt{(1-\rho)m_X\alpha}$$
$$\overset{\text{Lemma 14}}{\leq} \Xi\sigma_k$$

where the second inequality follows from the assumption. $\square$

We can now state the regret bounds by combining this result with Lemma 13.

### B.6.1 Noiseless Case

**Proof** (Theorem 4) If we set $C = 1/2$ in Lemma 13 we see that the to have precision $\eta$ and a guarantee that $\|\hat{g}\|_{\mathcal{H}_k}^2 \leq 2\|g\|_{\mathcal{H}_k}^2$ we will need the following to hold

$$\underbrace{\|X - \hat{X}^{(k)}\|_F}_{\tau} \leq \underbrace{\min\{1/2, \frac{\eta}{\eta + 2C_2\sqrt{k}(1+\bar{\varepsilon})}\}}_{\Xi} \sigma_k$$

The result will follow from picking $m_X, m_\Phi$ and $\varepsilon$ in Lemma 17 so that the above holds and the sampling complexity is $m_X(m_\Phi + 1) = \mathcal{O}^*(\sqrt{T})$. We will pick $m_X$ as in proof of Lemma 16. Now, the inequality above will hold if

$$m_\Phi \geq \Xi^{-2} \frac{1.44 C_2 k^5 \varepsilon^2 d^2}{(1-\rho)\alpha} m_X$$

$$\Longleftarrow$$

$$m_\Phi \geq \Xi^{-2} \frac{1.44 C_2^3 k^6 \varepsilon^2 d^2}{(1-\rho)\alpha^2\rho^2} \log(3/\delta)$$

If $\Xi = 1/2$ then the claim can be easily proven as the dependence on $T$ is very weak. Otherwise the above inequality transforms to (setting $\eta = T^{-1/2}$)

$$m_\Phi \geq (1 + 2\sqrt{k}TC_2(1+\bar{\varepsilon}))^2 \frac{1.44 C_2^3 k^6 \varepsilon^2 d^2}{(1-\rho)\alpha^2\rho^2} \log(3/\delta)$$

Which is satisfied by setting for some constant $C$ depending on $\rho, C_2$ and $\bar{\varepsilon}$

$$m_\Phi \geq C \frac{k^{6.5} \varepsilon^2 d^2}{\alpha^2} T \log(3/\delta)$$

We can now apply the fact that $\alpha = \Theta(1/d)$. Hence, the claimed guarantees will hold if we pick $m_\Phi = \mathcal{O}(k^2 d \log(1/\delta))$ and $\varepsilon = \frac{1}{k^{2.25} d^{3/2} T^{1/2}}$. We finally apply Lemma 1. $\square$

### B.6.2 Noisy Case

The proofs are similar to the previous case, we just have an additional term that we have to bound in Lemma 17. We do this by resampling which will reduce $\sigma$.

**Proof** (Theorem 5) Unfortunately for this case we can not achieve cumulative regret of $\mathcal{O}^*(\sqrt{T})$. Let us try to see how many samples do we need to have $\eta = \mathcal{O}(T^{-\beta})$ for some $\beta \in (0, 1)$. We will pick $m_X$ the same way as in Lemma 16. We will average out each point $m_\Phi n/\varepsilon^2$ times in which case in order the condition in Lemma 17 to hold we need

$$\sqrt{4.8k}\left(\frac{C_2 \varepsilon d m_X k^2}{2\sqrt{m_\Phi}} + \frac{6\sqrt{m_X}\sigma}{\sqrt{n}}\right) \leq \Xi\sqrt{(1-\rho)m_X\alpha}$$

Following the proof from the noiseless case we can make

$$\sqrt{4.8k}\frac{C_2 \varepsilon d m_X k^2}{2\sqrt{m_\Phi}} \leq \frac{1}{2}\Xi\sqrt{(1-\rho)m_X\alpha} \tag{6}$$

by choosing $m_\Phi = \mathcal{O}(k^2 d \log(1/\delta))$ and $\varepsilon = \frac{1}{k^{2.25}d^{3/2}T^\beta}$. We will now assure that

$$\sqrt{4.8k}\frac{6\sqrt{m_X}\sigma}{\sqrt{n}} \leq \frac{1}{2}\Xi\sqrt{(1-\rho)m_X\alpha} \tag{7}$$

which is satisfied if for some $C$ depending on $\rho$

$$n \geq \Xi^{-2}\frac{Ck\sigma^2}{\alpha}$$

Again the case $\Xi = 1/2$ easily yields the claimed bounds as then $\Xi$ is independent of $T$. Otherwise we need

$$n \geq (1 + 2\sqrt{k}T^\beta C_2(1 + \bar{\varepsilon}))^2 \frac{Ck\sigma^2}{\alpha}$$
$$\Longleftarrow \text{ For some } C' \text{ depending on } C_2, \rho$$
$$n \geq C'\frac{k^2\sigma^2}{\alpha}T^{2\beta}$$

Hence, as $\alpha = \Theta(1/d)$ we have to resample $\mathcal{O}(k^2 d\sigma^2 T^{2\beta} m_\Phi/\varepsilon^2)$ times. By combining Equations 6 and 7 and applying Lemma 17 we can achieve $\eta = \mathcal{O}(T^{-\beta})$ and also have $\|\hat{g}\|_{\mathcal{H}_k} \leq 2\|\hat{g}\|_{\mathcal{H}_k}$ using a total of $m_X m_\Phi^2 n/\varepsilon^2 = \mathcal{O}(k^{11.5}d^7\sigma^2 T^{4\beta}/\varepsilon^2 \log(1/\delta)^3)$ samples. To bound the regret we use Lemma 1 and we will get both $T^{1-\beta}$ and $T^{4\beta}$ as terms in the bound. If we optimize for $\beta$ we get $\beta = \arg\min_\beta\{1 - \beta, 4\beta\} = 1/5$ and this proves the claim. $\square$

# C   Optimizing the Branin Function

(a) Branin - Average regret.

(b) Branin - Simple regret.

(c) The Branin function.

The Branin function is a classical benchmark for global optimization. The function is two-dimensional and we hide it in a random two-dimensional subspace in 100 dimensions. It is flat near the origin and peaks very sharply on two opposite corners of the square. The heights of these peaks differ, which makes it even more complicated to optimize. To encourage more exploration we have increased $\beta_t$ by a factor of 15. This objective is very challenging because it is not well modeled with a GP prior and it is relatively flat near the origin, which means that it does not satisfy our assumptions. For example we see that even when we know the subspace exactly GP-UCB does not perform well. Moreover, its maximum is concentrated at the boundary of the region – hence we would need an extremely good approximation to the subspace in order to be able to achieve low regret.