[Reviews · NeurIPS 2013]

Submitted by Assigned_Reviewer_5

This is a very interesting and well written paper. It further the study of GP bandits initiated in Srinivas et al. Precisely in the present paper the authors assume that the unknown function only varies along a few dimensions k << d. They show regret bounds that are exponential in k (as expected) but only polynomial in d. I think that this is an important contribution.

Minor comments:
- the comment right before Section 5 on the mismatch of the effective dimension is very important. It is definitely worth exploring the effect of underfitting the value of k
- you could perhaps provide a reference for the notion of simple regret as it is not so well known.
Summary: Important contribution to GP bandits.

Submitted by Assigned_Reviewer_9

This paper considers optimization of high-dimensional expensive-to-compute
non-convex objective functions, under the assumption that the high-dimensional
function only varies in a low-dimensional subspace. The paper proposes an
algorithm with two steps. The first step is to estimate the low-dimensional
subspace, and the second step is to apply the GP-UCB algorithm to the estimated
low-dimensional objective function.

Overall, I think the paper is interesting, and applies a reasonable idea to an
important and difficult problem. I did however, see a number of practical and
algorithmic drawbacks to the proposed approach, some potential issues with the
theoretical analysis, and some weak points in the numerical experiments. These
are described in more detail below.


1. Although the method is interesting, there are several drawbacks in its design:
a. The method does not improve its initial estimate of the subspace while in
the second optimization stage.
b. It requires knowing several constants which would be hard to know in practice.
k, the number of dimensions in the low-dimensional subspace
B, an upper bound on the RKHS norm of the function being optimize
sigma, the variance of the noise
c. It requires making some assumptions about the underlying function g (in C^2
with Lipschitz continuous second order derivatives, and a full rank Hession at
0). It is claimed that these assumptions are "satisfied for many functions",
which is true, but in practice, for a given high-dimensional non-convex
expensive-to-compute function to which we want to apply this algorithm, I think
it is unlikely that we would know whether or not that function satisfied these
assumptions. It also requires assuming that the noise is Gaussian with
constant variance. Though these assumptions may be unavoidable, it would be
better to do numerical experiments to investigate whether failing to meet the
assumptions causes the behavior of the proposed algorithm to decay
significantly.
d. Since \hat{\sigma} depends on T (see point below) and GP-UCB is run with
parameter \hat{\sigma}, we must decide in practice how many iterations we will
run this algorithm before we start running it. If we decide later that we want
to run more samples later, we have to restart from scratch or lose our
theoretical guarantee.
e. The proposed algorithm does not do very well on simple regret, as the very
simple algorithm RandomH-UCB outperforms it on two out of three problem settings.
f. There is no way to include prior information from the practitioner about
which subspaces are likely to be important. In my practical experience with
these kinds of problems, it is common that the person close to the application
has a good idea of which coordinate directions are likely to matter, and which
are not. It is only on the margins where they are uncertain. (It is true that
it is more difficult to ask a practitioner which subspaces, not necessarily
aligned with coordinate axes, are likely to matter). Indeed, a good benchmark
algorithm for comparison would be to take a real problem, ask a person close to
the problem which 5 coordinate directions he thinks are most important, and run
GP-UCB on just those coordinate directions, picking arbitrary values for the
other coordinates.
g. The assumption is made that there is absolutely no variation in the function
outside of the low-dimensional subspace. In practice, I think this is unlikely
to be true. Instead, it seems more likely that there is a small amount of
variation outside of the low-dimensional subspace.

2. In the choice for \hat{\sigma} on line 213, T does not appear. Of course,
it should appear, since the probability that the maximum of T iid
normal(0,sigma) random variables exceeds any fixed threshold goes to 0 as T
goes to infinity. My understanding from reading the surrounding text is that
the omission of T is just a typo, but there appears to be an error in the
analysis later resulting from this omission (see point 3b).

3. Two potential issues with the proofs:
a. B is specified as an upper bound on the squared RKHS norm of the function
\hat{g}. But in the case with noisy measurements, \hat{g} is determined in
part based on the estimated subspace, which is random. So even though g is a
deterministic function, \hat{g} is random, and so its squared RKHS norm is
random. How do we know there exists an upper bound B that holds almost surely?
b. \hat{\sigma} seems to depend on T, although an apparent typo hides this
dependence (see discussion in other comment about line 213). Buried in the
constant in Theorem 3 of the GP-UCB paper [Srinivas et al. 2012] (upon which
the proof of Theorem 1 is based) is a dependence on \hat{\sigma}, which that
paper assumes to be a constant. This extra dependence on T should be included
in the bound on regret.

4. Theorem 5 uses g rather than \hat{g} in the term R_{UCB}(T,g,\kappa), unlike
Lemma 1, which used \hat{g}. I believe this is a typo. If it is not a typo,
how is moving from \hat{g} to g accomplished? The proof doesn't make this
clear. If it is simply a typo, this bound is not as explicit as we would like,
because R_{UCB}(T,\hat{g},\kappa) is a random variable that depends in a
complicated way on the algorithm, rather than depending only on constants.


5. I do not see precise guidance as to how to choose the number of times we
resample each point in practice, either in Theorem 5 or in the section of Noisy
Observations right before it. Only its big-O dependence on other quantities is
given.


6. The numerical experiments are not especially thorough.
a. They only perform 20 replications which leaves the standard error quite
wide. Indeed, is the halfwidth of the bars just a single standard error? If
so, it looks like the standard errors are large enough that we can't claim with
statistical significance that SI-BO is better than either RandomS-UCB or
RandomH-UCB in most cases. Or am I misinterpreting the figure?
b. The paper does not compare with the method from reference [19] in the
paper, Wang et al. 2013.
c. Only three test problems are investigated, (all of which are just benchmark
problems, and not especially interesting).
d. The values of k chosen are all really small, only 1, 2 or 3. I would think
that people would typically be interested in running this kind of algorithm
with slightly larger values of k.


Minor issues:

Pertaining to the paragraph in which line 213 appears, it is confusing that
here T is the number of GP-UCB iterations, but later, in Lemma 1, T is the
overall number of iterations.

The figure labels (a) and (b) in the supplement section C are reversed.

In the supplement, Lemma 1 is called Theorem 1.
Summary: Overall, I think the paper is interesting, and applies a reasonable idea to an
important and difficult problem. I did however, see a number of practical and
algorithmic drawbacks to the proposed approach, some potential issues with the
theoretical analysis, and some weak points in the numerical experiments.

Submitted by Assigned_Reviewer_10

This paper is about optimizing high dimensional noisy functions under the
assumption that the true function lives on a linear subspace of much lower
dimension. The algorithm pastes together an existing linear subspace
finder followed by a GP optimizer (GP-UCB). The contribution is to
theoretically analyze the combined algorithm and show that it is polynomial
in d, the ambient dimension.

The paper is well written. The problem is one of significant interest, and
the resulting polynomial regret bounds are nice. The paper is also nice to
read because there is plenty of fodder to think about how to solve this
problem.

My enthusiasm about it is reserved for several reasons. Mainly, I find it
hard to believe this is the algorithm one would really use in practice on
this problem. Although its nice to be polynomial, I doubt that results
with a factor of k^{11.5} d^7 provide non-trivial bounds compared to what
you would get by bounding the function overall and looking at the worst
possible regret from the upper and lower bounds of the function. This
leads me to question whether the theoretical bounds provide any real
insight to whether this is a good algorithm. My concerns are somewhat born
out in the empirical results. The strawmen are optimizing in the original
space and optimizing in a randomly chosen subspace of the correct
dimension. If one believes this is a challenging problem in the first
place (I do) then one would expect a good algorithm to crush these weak
alternatives. The results, however, show the wins to be quite modest. And
working in the original space actually gets the crushing victory on one of
the data sets when the performance metric is simple regret.

What algorithm to use in practice is an interesting question. At least,
one would hope to keep refining the estimate of the correct subspace using
samples taken during the optimization phase.

Some smaller comments:

line 55: is is
line 106: our this
line 125: only only
footnote 2: is a

the reference to footnote 4 appears at the second use of O^* rather than
the first.

the choice of curves in figure 1 right is a little unfortunate. optimizing
the curve with cosine similarity .869 would actually yield a poor value of
the true function while optimizing the one with a cosine similarity of
0.216 would turn out fine.

the connection between the paragraph on mis-specified k and fig 3 b,e is
not given. either the text should point there, or the caption should say
what each subfigure is.
Summary: A nice paper with some theoretical results but someone actually trying to solve this problem would probably do something different.
Author Feedback

Author rebuttal: We would like to thank the reviewers for their insightful comments. Please find our response below.

* We do not need to know the variance of the noise exactly, but only have an upper bound on it. Similarly, we need a bound on the RKHS norm, which can be estimated using the “doubling trick” - keeping an estimate of it and doubling it based on observations.
* There was indeed a typo in how \hat\sigma is defined: it should be chosen as \sigma * \sqrt{ 2 log 1/delta + 2 logT + log (1 / 2\pi) }.
* Reviewer #9 is right that the bounds in Theorem 3 in [Srinivas et al.] depend on \hat\sigma. In order to make the dependence on \sigma instead of \hat\sigma, the term C_1 will become 8 / log(1 + 1 / \sigma^2 (2 log(T/\delta) + log(1 / 2\pi)). Note that the function u(x)=1/log(1+1/log(x)) is a very slowly growing function, for example u(10^20) ≈ 46.5.
* Regarding the relation between the RKHS norms of g and \hat{g}: This is not a typo and we have a dependence on the RKHS norm of the true function g. We achieve this by showing that the norm of \hat{g} is only within a constant factor of the norm of g. We do not claim that this holds a.s., but that it holds with probability at least 1-\delta. You can take a look at Lemma 9 in the supplementary material, which says that the norm of \hat{g} is close to the norm of g if |det M| is bounded away from 0. Then, we show that using the subspace learning method that we use, this is indeed true w.p. at least 1-\delta. This results in Lemma 13, which is then used to show the regret bounds.
* The number of times resampling should be done is given in a big-Oh form due to the fact that we use only the asymptotic behavior of \alpha. To get more exact bounds, you can take a look at the cited paper by Tyagi and Cevher from NIPS2012 that analyzes the behaviour of \alpha.
* We leave the analysis for mis-specified k and and functions that also have some variation along the other dimensions for future work.
* We focused mainly on the cumulative regret and showed the simple regret for comparison.